

# Exploring micro-scale heterogeneity as a driver of biogeochemical transformations and gas transport in peat

Lukas Kohl[1,2,3], Petri Kiuru[4], Marjo Palviainen[5], Maarit Raivonen[1], Markku Koskinen[2,6], Mari Pihlatie[2,6], and Annamari Laurén[4,5]

[1]Institute for Atmospheric and Earth System Research (INAR)/Physics, Faculty of Science, University of Helsinki, Helsinki, Finland
[2]University of Helsinki, Faculty of Agricultural Sciences, Department of Agricultural Sciences, Helsinki, Finland
[3]Department of Environmental and Biological Sciences, Faculty of Science, Forestry and Technology, University of Eastern Finland, Kuopio, Finland
[4]School of Forest Sciences, Faculty of Science, Forestry and Technology, University of Eastern Finland, Joensuu, Finland
[5]Department of Forest Sciences, University of Helsinki, Helsinki, Finland
[6]Institute for Atmospheric and Earth System Research (INAR)/Forest Sciences, Faculty of Agriculture and Forestry, University of Helsinki, Helsinki, Finland

**Correspondence:** Lukas Kohl (lukas.kohl@helsinki.fi)

**Abstract.** Peat pore network architecture is a key determinant of water retention and gas transport properties, and has therefore been hypothesized to control redox conditions in and greenhouse gas emissions from peat soils. Yet, experimental proof of the impact of the pore network structure on biogeochemical reactions remains scarce. Here, we report on a $^{13}$C pulse-chase assay developed to functionally explain and visualize the cm-scale heterogeneity in greenhouse gas emissions in peat cores.

We injected a $^{13}$C labeled substrate ($^{13}$C$_2$-acetate) at different depths in the peat cores and monitored its conversion into $CO_2$ and $CH_4$ and the subsequent transport to the core headspace. We then measured the pore network architecture of the same cores by X-ray microtomographic imaging and constructed the air-filled pore networks using pore network modeling. We found large heterogeneity among the replicate cores and injections, indicating the effects of cm-scale heterogeneity on biochemical processes and gas transport. This heterogeneity was largely present at the core (10 cm) and within-core (cm) scale

heterogeneity whereas little additional variance occurred on the stand (>10m) scale. Deeper injections resulted in a smaller faction of the label being converted to $CO_2$ and this fraction being emitted more slowly from the peat cores. Greater peat air-filled porosity was and pore network metrics could not explain the fraction of label converted to $CO_2$, but greater porosity as well as higher clustering coefficients and betweenness centrality were associated with slower $CO_2$ emissions.

## 15   1   Introduction

Peat pore network architecture controls microscale gas exchange, which determines redox conditions, the production of the greenhouses gases carbon dioxide ($CO_2$) and methane ($CH_4$), and their transport by diffusion and ebullition (Kiuru et al.,



2022b; Ramirez et al., 2016). Yet, empirical methods that explain and visualize the role of pore networks and small-scale heterogeneity in the regulation of soil functions remain elusive.

Peatlands are of global importance as modulators of biogeochemical cycles and greenhouse gas balances (Gorham, 1991; Limpens et al., 2008). Globally, more than 600 Gt of C are stored in peat layers (Yu et al., 2008), which are sensitive to drainage, forest management, and changes in environmental conditions. In a warming climate, peatlands are becoming a major source of greenhouse gases (GHG) such as $CO_2$ and $CH_4$ (Leifeld et al., 2019; Frolking et al., 2011). In peat, the production of $CO_2$ and $CH_4$ are primarily determined by soil temperature and oxygen ($O_2$) supply (McCarter et al., 2020). Where sufficient $O_2$ is

available, heterotrophic respiration dominates and peat is decomposed to $CO_2$. In the absence of $O_2$, peat decomposition uses other electron acceptors, which eventually leads to methanogenesis. This occurs, for example, below the water table (WT) and above the WT in aerobic microsites (anaerobic pockets) (Wachinger et al., 2000; Hagedorn et al., 2011). At this micro-scale, $O_2$ concentrations depend on the balance between $O_2$ consumption, driven by temperature and substrate availability, and on the $O_2$ transport from the atmosphere to soil. This transport, in turn, depends on the peat water content and the connectivity and

structure of the air-filled macropore network in the peat (Kiuru et al., 2022a, b). Small-scale heterogeneity in the pore structure may explain the noisy and peaky patterns of methane emissions typically observed in field conditions (Xu et al., 2016).

Air-filled macropores in peat can be identified by microtomographic imaging that are abstracted into pore network models (PNM) (Kettridge and Binley, 2011; Rezanezhad et al., 2016; Kiuru et al., 2022a, b). This approach allows simulation of physical processes, such as water retention (Kiuru et al., 2022a) and gas transport (Kiuru et al., 2022b). Pore networks can also

be analysed based on network metrics (Newman, 2006), and pore network imaging can help in identifying macropores that are isolated from the larger pore network and from the atmosphere (Kiuru et al., 2022a). PNM has suggested that the proportion of such isolated pores decreases with decreasing water content, and that hysteretic behaviour of soil wetting and drying leads to greater abundance of isolated pores during the drying than during imbibition (wetting) (Kiuru et al., 2022a).

Despite these progresses in pore network modelling, experiments that demonstrate how pore networks regulate production of

$CO_2$ and $CH_4$, remain missing. One significant reason for this knowledge gap is the lack experimental approaches to localize biochemical reactions within intact peat cores. Here, we aim to demonstrate the microscale heterogeneity of both pore networks and biogeochemical processes within peat cores. To achieve this, we injected an isotopically labelled substrate ($^{13}C_2$-labelled sodium acetate, $^{13}CH_3COONa$) and followed the emissions of $^{13}CH_4$ and $^{13}CO_2$ from these cores during heterotrophic respiration (Reaction 1) and acetoclastic methanogenesis (Reaction 2). Note the position-specific conversion of $C_2$-carbon to

methane in R2.

$$^{13}CH_3COOH + 2O_2 \rightarrow\, ^{13}CO_2 + CO_2 + 2H_2O \tag{R1}$$

$$^{13}CH_3COOH \rightarrow\, ^{13}CH_4 + CO_2 \tag{R2}$$

We compared the effect of injections at different depths and compared wetting and drying peat cores at the same water potential on the conversion rate of the injected label into $CO_2$ and $CH_4$ as well as the time lag between the injection and the emission



of these gases from the top of the peat core. After the manipulation experiment, we conducted microtomographic imaging and analysed the pore space above the injection depth. We hypothesize that greater air-filled porosity would be associated with a higher conversion of the methyl group of acetate to $CO_2$, less conversion to $CH_4$, and a more rapid onset of emissions.

## 2 Methods

### 2.1 Site description and peat sampling

Peat samples were collected from a forest (60°38′N, 23°57′E, Lettosuo, Tammela) in Southern Finland in December 2021. The site was drained in 1969 with parallel ditches in 40 m spacing. The mean annual temperature and precipitation at Lettosuo are 5.2 °C and 621 mm (Jokinen et al., 2021). The peat type iss Carex peat. The site was originally a mesotrophic fen classified as a herb-rich tall-sedge birch–pine fen (Laine and Vasander, 1996). The forest stand is dominated by Scots pine (*Pinus sylvestris* L.) and downy birch (*Betula pubescens* Ehrh.) with an undergrowth composed of Norway spruce (*Picea abies* Karst.). The dominant height of the stand was 20 m and volume of the growing stock was 230 m3 ha-1. Ground vegetation consists of dwarf shrubs (coverage 4 %) including *Vaccinium myrtillus* L. and *V. vitis-idaea* L., as well as herbs (coverage 10.6 %). A detailed site description is available in Kiuru et al. (2022a).

Peat core samples were collected from seven replicate pits that were located at least 30 m apart from each other. The cores were collected by removing the top 15 cm of soil, including a thin ice wedge that had formed in this layer. At each pit, two parallel samples were extracted into cylindrical cores (10.0 cm height, 10.0 cm diameter) from the depth of 15-25 cm.

### 2.2 Sample storage and water potential setup

Samples were wrapped in shrink-wrap foil and stored at +4 °C until the pretreatment, where all samples were water saturated and placed on two sand beds that were hydraulically connected to hanging water columns (Eijkelkamp sand bed). One parallel sample from each pit was retained close to saturation (5 cm water column below the mid-point of the sample corresponding to a water potential of -5 hPa) while the other parallel sample was moderately drained (35 cm water column, i.e., -35 hPa). All samples were then set to a water potential of -20 hPa; consequently, one parallel sample of each pair reached the final water potential during drying and the other parallel sample during wetting.

### 2.3 Measurement setup

For measurements, the peat samples were equipped with ca 7 cm high collars made from 5 mm thick neoprene rubber sheets that were outfitted with two ports for polytetrafluroethylene (PTFE) tubing. The bottom of each core and the top of each collar were sealed with shrink-wrap foil secured with rubber rings. One of the tube ports was connected to a 16-port selector valve (VICI model EMT-STF16MWE), and further to a Picarro G2201-i ($^{13}CO_2/^{13}CH_4/H_2O$) as well as a parallel pump for increasing the flow rate through the measurement system (Fig. 1). The other tube port was equipped with a 1 m long tube open





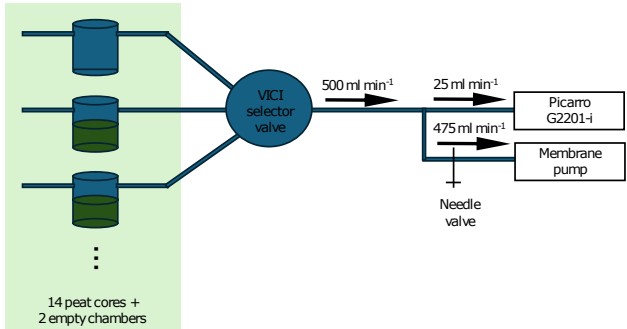

**Figure 1.** Depictions of the measurement setup. Ambient air was pulled through a headspace chamber to a Picarro G2201-i cavity ring-down spectroscopic $^{13}CO_2/^{13}CH_4$ analyser. A parallel line to additional membrane pump was used to increase the sample flow rate and regulate it with a needle vale. The system was connected to 16 chambers (14 peat cores and two empty chambers) using a VICI 16-port selector valve. Headspace air from each chamber was analyzed for 10 minutes once every 160 minutes, and $CO_2$ and $CH_4$ emitted by the peat cores was allowed to accumulate in the chamber headspace for 150 minutes between measurements.

to the atmosphere. The total flow rate of the system was set to 500 mL $min^{-1}$ by regulating the air flow to the auxiliary pump

using a needle valve. In addition to the 14 peat samples, two empty chambers were included in the system as blank controls.

The measurement system was set up to pull air sequentially from each chamber for 10-minute periods. Each chamber was analysed once every 160 minutes, with a 150-minute break between the measurements, during which $CO_2$ and $CH_4$ were allowed to accumulate in the chamber headspace. The time period between the two consecutive air pulling events in a sample is called hereon as a "closure". During the measurement event, the analyser initially measures the concentration and isotope

values of $CO_2$ and $CH_4$ accumulated in the headspace since the previous measurement event of the chamber, followed by increasing dilution of the headspace with ambient air (Fig 2). After ca. 2-3 minutes, a dynamic equilibrium is reached where the headspace $CO_2$ and $CH_4$ concentrations equal the concentration in ambient air plus the current chamber emissions. The closure times of the two empty chambers were slightly different (500 and 700 seconds) for easier identification of the chambers in the raw data.

## 2.4 Labeling experiment

$^{13}$C-labeled substrate was injected three times into each peat sample with 7 days intervals between injections. We injected 1 ml of 10 mM $^{13}C_2$-sodium acetate solution (i.e., a total of 10 $\mu$mol label per sample) followed by 1 mL ultrapure water. The injections were applied using syringes and hypodermal needles at 2.0, 5.0, and 8.0 cm depth in a different order for each parallel sample.

After each injection, the needles were closed using 3-way valves to prevent gas exchange through the needle and left in the peat core for the rest of the experiment. After dismantling the experiment, the needles were removed to avoid metal objects interfering with microtomographic imaging and wooden toothpicks were inserted into the vacated needle canals to mark the



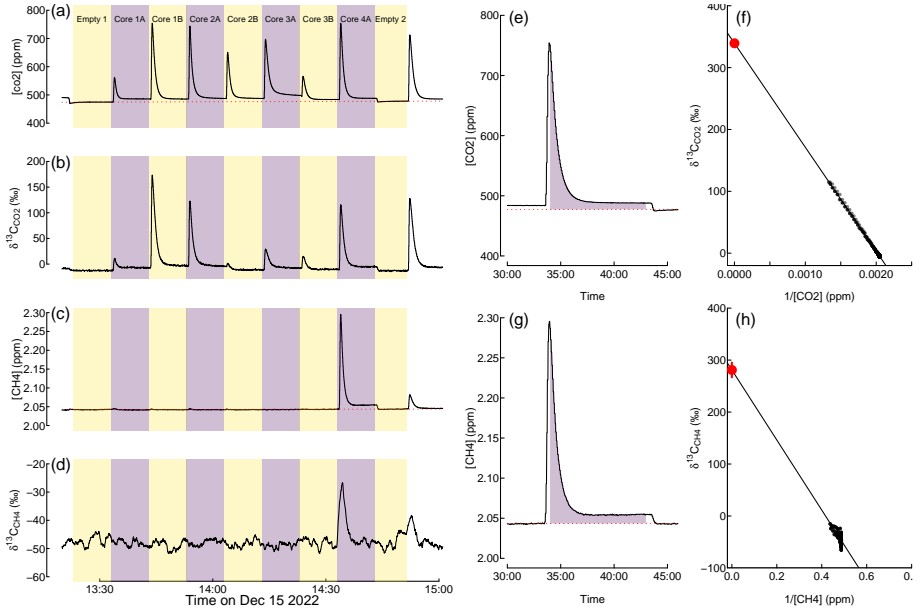

**Figure 2.** Example of raw data, including measured $CO_2$ and $CH_4$ concentrations (a, c), measured carbon isotope ratios ($\delta^{13}C$) in $CO_2$ and CH4 (b, d), integrated area for calculating $CO_2$ and $CH_4$ emissions rates (e, g), and Keeling plots for estimating the $\delta^{13}C$ value of peat-emitted $CO_2$ and $CH_4$ (f, h). The dashed red line in (a), (c), (e), and (g) represents the concentration baseline, which was estimated by interpolation from empty-chamber measurements. In (f) and (h), black symbols represent measured data points, solid lines represent linear regressions, and read points and error bars indicate the $\delta^{13}C$ value of peat-emitted $CO_2$ and $CH_4$ (i.e., the intercept of the regression line) and its 2 standard error uncertainty.

position of the injections in the $\mu$CT image. However, the positions of these could not be identified in the $\mu$CT images, preventing the identification of the exact injection location in pore networks.

## 2.5 Flux calculations

For each chamber closure, we calculated the amount $CO_2$ and $CH_4$ emitted during a closure from the measured gas concentration using Eq. 3 after subtracting a baseline concentration determined by linear interpolation between the two closest blank measurements (Figs 2a, 2c, 2e, 2g). The emission rates (in mol $min^{-1}$) were then calculated by dividing the amount of accumulated gas (in mol) through the time between measurements (160 minutes; Eq. 1).

$$F = \frac{A \cdot f}{V_{mol} \cdot t_{cycle}} \tag{1}$$

Where, $F$ is emission rate (mol $CO_2/CH_4$ $min^{-1}$), $A$ is the integrated baseline-corrected gas concentration from the maximum mixing ratio to 30 seconds before the end of the closure (mol $CO_2/CH_4$ $mol^{-1}$ min), $f$ is the gas flow rate (0.5 L $min^{-1}$),



$V_mol$ is the molar volume of an ideal gas (24.055 L mol$^{-1}$ at 20C and 101.325 hPa), and $t_cycle$ is the length of a measurement cycle (160 min).

We further calculated the fraction of peat-derived $CO_2$ and $CH_4$ $f_i$ at each measurement time point $i$ according to eq. 2 (in the case of net emissions) or eq. 3 (in the case of net uptake), where $c_i$ stands for the concetration at time point $i$, $c_{bg}$ for the concentration in ambient air derived from empty chamber measurements, and $c_min$ and $c_max$ for the lowest and highest concentration during the measurement period.

$$f_i = \frac{c_i - c_{bg}}{c_{max}} \tag{2}$$

$$f_i = \frac{c_{bg} - c_i}{c_{min}} \tag{3}$$


Further, we then calculated the $\delta^{13}C$ values of $CO_2$ and $CH_4$ emitted during each closures by mass balance as the the intercept of the linear regression between the measured $\delta^{13}C$ values and $1 - f$ (**Figs 2f, 2h**). We converted the $\delta^{13}C$ values to atom percent excess (APE) according to Eq. 4, where $\delta^{13}C_{sam}$ is the measured $\delta^{13}C$ value, $\delta^{13}C_{cont}$ is the $\delta^{13}C$ value of an unlabelled sample, assumed -28 ‰for $CO_2$ and -70‰for CH4, and $R_ref$ is the absolute $^{13}C/(^{12}C+^{13}C)$ ratio of the $\delta^{13}C$ reference material (VPDB; 0.01111233).


$$APE = \frac{\delta^{13}C_{sam} - \delta^{13}C_{cont}}{1000} \cdot R_{ref} \cdot 100 \tag{4}$$

The rate of label-derived $CO_2$ and $CH_4$ ($F_L$, mol min$^{-1}$) emissions were calculated based on the total emission rate $F$ and the $APE$ measured during each closure (Eq. 5).

$$F_L = F \cdot APE/100 \tag{5}$$

Eqs. 4 and 5 were also applied to quantify emissions of label-derived $^{13}CH_4$ from peat cores that showed net uptake of (unlabelled) $CH_4$. In that case both $F$ and $APE$ are negative, resulting in a positive $F_L$.

To correct for carry-over from one injection to the next (e.g. emissions of $^{13}CO_2$ and $^{13}CH_4$ derived from the first injection after the second injection) we fitted an exponential decay function to the $^{13}CO_2$ and $^{13}CH_4$ emissions rates over the four days prior to the next injection. This curve was then extrapolated to the measurement period after the subsequent injections and 130 subtracted from the observed emissions.

## 2.6 Three-dimensional $\mu$CT imaging and image processing

After the labeling experiment, the peat samples were covered with shrink-wrap foil and stored in +4 °C until $\mu$CT imaging with a GE Phoenix Nanotom system. The 16-bit 3D grayscale images obtained in the $\mu$CT reconstruction had size of 1268 by 1120 by 1120 voxels (cubic 3D image element) at 100 µm resolutions.





In the image preprocessing stage, the 3D grayscale images were converted to 3D binary images that separated void (air) voxels from voxels representing solid space and water using the Python image processing packages scikit-image(Van Der Walt et al., 2014) and SciPy ndimage (Virtanen et al., 2020) and the image analysis toolkit PoreSpy (Gostick et al., 2019). First, the 3D grayscale images were straightened and cropped to a size of 1000 by 900 by 900 voxels according to the inner dimensions of the cylindrical tubes. A cylindrical peat volume with a height of 1000 voxels and a diameter of 900 voxels was further

selected using PoreSpy. Before the noise filtering and binary segmentation stages, the images were linearly mapped to an 8-bit representation. The mapping interval extended from 0.5 % to 99.5 % of the cumulative image gray-level intensity distribution so that the long tails of the intensity distribution formed by noise or occasional small mineral grains were removed. The 8-bit images were then noise-filtered using a 3D median filter with a 2-voxel radius. Finally, the images were segmented into void and solid volumes with the global Otsu thresholding algorithm (Otsu, 1979). Isolated solid regions were removed from the

resulting binary images using a method for the determination of disconnected voxel space in PoreSpy.

## 2.7   Image analysis

Because the samples had shrunk slightly and their top and bottom surfaces were rough and uneven, the sample images were also cropped in the vertical direction so that the final image domain did not contain any external void space. The height of the final cylindrical domain with a diameter of 90 mm varied from 75 mm to 95 mm. The air-filled porosity of each image domain

was calculated as the ratio of the number of void voxels to the number of total voxels in the domain. The vertical air-filled porosity distribution was obtained by determining the void-voxel ratio for each horizontal voxel layer. For the determination of the radial air-filled porosity distribution, the domain was divided into 45 hollow cylinders with equal diameter increments. Because the samples had shrunk in the vertical direction, some void space had been generated between the peat matrix and the tube walls. To only include the internal void space of the samples, the vertical porosity distribution was calculated for a

cylindrical domain with a diameter of 80 mm.

## 2.8   Pore networks

Pore networks were extracted from the final cylindrical domains of the binary images using a marker-based watershed segmentation method (Gostick, 2017). The segmentation algorithm divides the void space into individual pore regions and determines the connections between the pores and the locations of the two-dimensional interfaces between neighboring pores called pore

throats. Because the feature resolution of a $\mu$CT-derived image is generally approximately twice the image voxel size (Stock, 2008; Elkhoury et al., 2019), the size of the smallest distinguishable feature in the images was 200 $\mu$m.

    The pore system generated by the extraction algorithm was divided into clusters of interconnected pores and a group of single isolated pores using the open-source pore network modeling package OpenPNM (Gostick et al., 2016). The largest of these clusters, which was assumed to be the only cluster that extends through the network domain in the axial direction and

which was therefore the relevant space regarding gas transport through the domain, was defined as the pore network. The pore volume was determined by counting the number of voxels in an individual pore region. Network porosity was defined as the ratio of the sum of the volumes of the pores in the network to the total volume of the domain. Further network metrics were



calculated following Kiuru et al. (2022a). Briefly, coordination number is defined as the average number of connections of each pore to other pores. Clustering coefficient as the probability that two pores connected to a given pore are also connected to each other. Closeness centrality is the reciprocal of the average shortest path length from one pore to each other pore in the network. Geometrical tortuosity and betweenness centrality represent transport properties of the pore network a certain direction (between top and bottom of the peat core) and as a whole.

## 2.9 Statistical analysis

We tested for treatment effects on parameters discribing $CO_2$ and $CH_4$ emissions by applying mixed effect models. We applied *injection depth*, *injection round*, and *moisture treatment* as fixed effects and *soil pit* as a random effect. For fixed effecst that significantly affected the dependent variable, we conducted estimated marginal means to identify significant differences between variable levels. In addition to these models, we tested for linear correlations between pore network measures and $CO_2$ flux parameters. All statistical analysis was conducted in the statistical programming environment R version 4.2.1 (R Development Core Team, 2015) using the *lme4*, *lmerTest*, and *emmeans* packages.

## 3 Results and discussion

### 3.1 Microtomography and pore architecture

The mean air-filled porosity in the 14 peat samples ranged 0.20 to 6.75% (average: 2.56%, standard deviation 2.02 %). Microtomographic imaging revealed high heterogeneity both within and between the peat cores. Four examples of vertical cross sections through the cores are shown in Figs. 3 and S1. Visual inspection showed large, mainly horizontally-oriented macropore systems in a dense matrix (Figs 3a,3c,3d), and vertically connected pore networks (Fig 3b) reflecting a looser peat structure. We found a large degree of vertical heterogeneity in air-filled pore-volume, originating from layered, horizontally oriented macropores (Fig. 3e) and air-filled cavities in the peat samples. In contrast, all peat cores show the same radial porosity trend from the center to the edge (Fig 3f). This indicated the absence of distinct vertical pore structure, which would be visible distinct features in these plots. All samples showed a similar increase of air-filled porosity towards the edge of the sample, an artifact of shrinkage caused by drying.

The visible structure of the peat reflects the original plant residues that formed the peat at the site as well as the changes during over time and the effects of site drainage. In forested peatlands the peat typically contains woody plant fragments and Carex residues, as is the case at Lettosuo. Woody fragments in peat increase spatial heterogeneity with large macropores compared to the more fine-pored and homogeneous sphagnum-derived peat (McCarter et al., 2020). The presence of dwarf shrubs likely also introduced looser peat structure and larger macropores. Site drainage enhances peat decomposition, which leads to a loss in macropore space and increases peat bulk density, particularly in the top layer of the peat (Minkkinen and Laine, 1998). Concurrent with the enhanced decomposition and compaction, a raw-humus layer forms on top of the peat. The



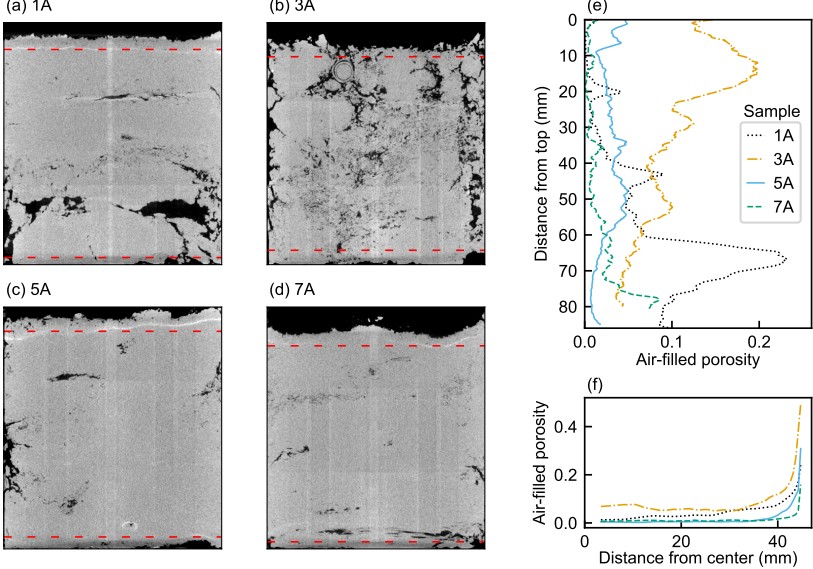

**Figure 3.** Axial cross sections of noise-filtered 3D $\mu$CT images of peat samples (a) 1A, (b) 3A, (c) 5A, and (d) 7A as well as vertical (e) and radial (f) profiles of the air-filled porosity of the samples. Red dashed lines in the images show the boundaries of the final network domain.

raw humus layer is mainly formed of litter originating from upland vegetation and might have influenced the underlying layer from which the samples in this study were collected, e.g. by forming a horizontally layered peat structure (Lauren, 1999).

Scanning whole peat cores required us to limit the measurement resolution to 200 $\mu$m. We thus captured only the largest of macropores. However, this resolution was sufficient to map air-filled pores in our samples, which were set to a water potential of –20 hPa. This corresponds to an equivalent pore size diameter of 150 μm. We were unable to identify the injection positions marked by wooden toothpicks used to mark them. We can therefore only compare GHG emissions to properties at the scale of the scanned peat cores, but not the local environment at the location of the injection.

## 3.2 $CO_2$ and $CO_4$ emissions from peat core

### 3.2.1 Background emissions of $CO_2$ and $CH_4$

All peat cores emitted $CO_2$ at a mean rate of 1.6±0.6 (1 SD among core) μmol h$^{-1}$ (range: 0.4 to 2.7 μmol hh$^{-1}$). These total emissions were not affected by the injections, as was indicated by the lack of differences of the background respiration after the injections at different depths (Fig 4a). We did, however, observe a trend towards higher $CO_2$ emissions after the third round of injections (Fig 4b) that approach significance (p = 0.076). This indicates the potential for a minor increase in peat respiration rates towards the end of the experiment. We observed no differences between the drying and wetting treatments (Fig 4c).

Three of the 14 peat cores acted as methane emitters with emissions rates up to 1.67 nmol h$^{-1}$, whereas the remaining 11 peat cores acted as small CH4 sinks with sink strength up to 0.05 nmol hh$^{-1}$. On average, peat cores were net emitters with



a mean flux of $0.32\pm0.95$ nmol CH4 hh$^{-1}$. All methane-emitting cores had low air-filled porosities (<1%), although not all
cores with low air-filled porosity emitted methane. We observed no significant changes in the background $CH_4$ emissions over
the course of the experiment and no difference in emissions after the injections at different depths (Fig 4d-4e). A trend towards
higher emissions in drying compared to wetting treatments (Fig 4f) was not significant (p=0.086).

The absence of methane emissions from most peat samples was consistent with the field environment where they were
collected: a drained peatland that currently acts as a net sink of methane (Korkiakoski et al., 2020). The water potential in our
experiment (-20 hPa) was comparable to the location of the water table (-40 to -30 cm) relative to the sampling depths. Our
results thus indicate the presence of individual methane emitting locations in a larger methane consuming stand.

The independence of background $CO_2$ and $CH_4$ emissions of injection depths indicates that the label (acetate) injections did
not affect the overall biogeochemistry of the peat cores (Fig. 4a, 4d). However, it is possible that the peat cores were affected
by higher temperatures, which may have increased the respiration rates towards the end of the experiment. Nevertheless, the
size of this disturbance was limited and acceptable in an experiment that was not designed to exactly replicate field conditions.

The trend towards higher methane emissions in the wetting compared to the trying treatment is interesting, as it indicates
higher methane emissions in peat cores that have been exposed to more oxic conditions prior to the experiment. This may have
been caused by the release of more labile substrates during aerobic episodes which can then be utilized by methanogens during
the following aerobic period.

### 3.3 Label-derived CO2 emissions

We followed the release of the label-derived $^{13}CO_2$ over 43-68 measurement cycles, that is, 114-181 hours. In these emissions,
we observed a high heterogeneity between the peat cores and in the response to individual injections (Fig. 5). Overall, we
observed the highest rates of $^{13}CO_2$ release over the first 24 hours after label injection (Figs. 5a-5c). However, only some of
the injections led to a strong, early $^{13}CO_2$ release. Other injections showed a longer response time lag, reaching maximum
$^{13}CO_2$ emission rates 24-72 hours after the label injection. Although this type of response typically showed lower maximum
emission rates (Figs. 5a-5c), it often reached a higher cumulative emission throughout the experiment (Figs 5d-5f).

To compare $^{13}CO_2$ emissions across experiments that had different runtimes, we integrated the observed emissions over
the first 41 measurement cycles (109.3 hours). Over this period, we found emissions ranging from 0.01 to 1.22 $\mu$mol $^{13}CO_2$
or 0.11 to 12.2% of the injected label. The average fraction of the label emitted as $CO_2$ decreased with injection depth, from
7.2% at 2 cm depth to 1.9% at 8 cm depth (F=12.2, p <0.001; Fig 6a). The emitted $^{13}CO_2$ did not differ between the injection
rounds or soil moisture treatments (Figs 6b, 6c).

To characterize the combined effect of the delayed onset of the label conversion to $CO_2$ and the diffusion time, we determined
the time from each label injection til half of the $^{13}CO_2$ emissions after the same injection had occurred ($t_{1/2}$). This level
was reached after 3 to 28 hours. Again, we found a significant difference between injections at different depths, with $^{13}CO_2$
emissions showing a greater average time lag at greater depths (10.5 hours at 2 cm depths vs. 17.1 hours at 8 cm depths).
Further, injection round or soil moisture treatment had no significant effect on $t_{1/2}$.



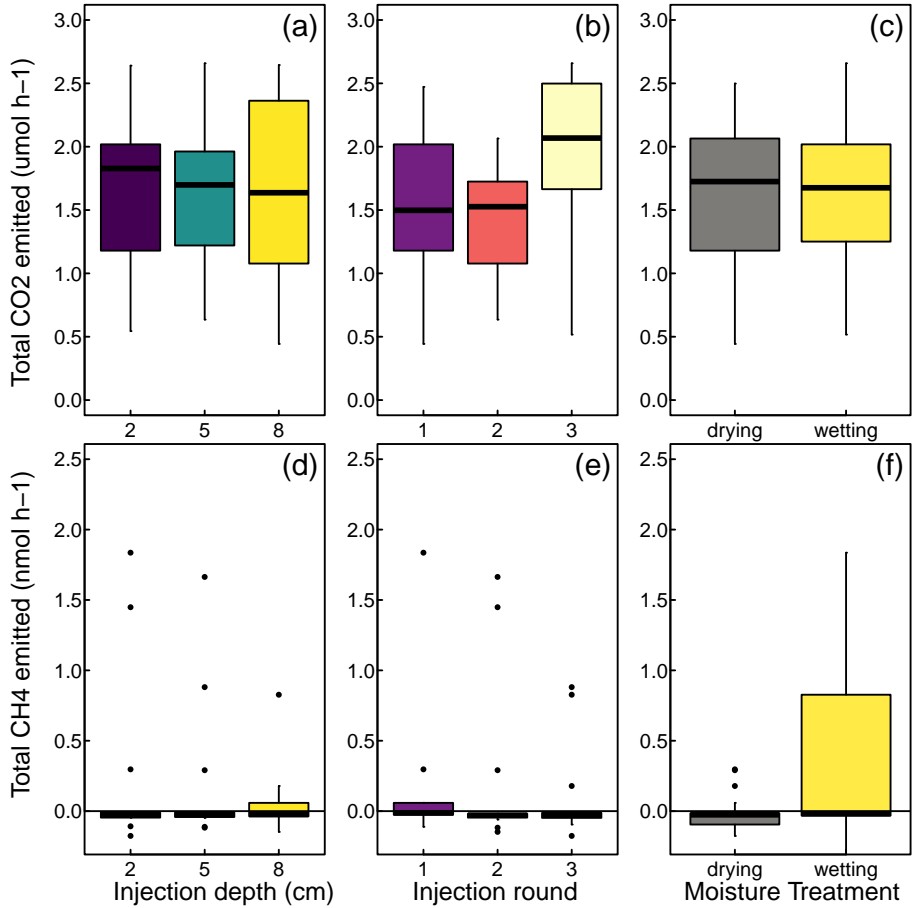

**Figure 4.** Total carbon dioxide ($CO_2$) and methane ($CH_4$) emissions from peat cores, that is, the sum of label-derived and non-label-derived emissions. Positive numbers indicate net release of gases to the atmosphere, negative numbers net uptake by the peat cores. Letters in panel (b) indicate significant differences between the injection rounds. No significant differences were found in any other case.

The slower or time-delayed emission of $^{13}CO_2$ after injections at greater depth could have been caused by several factors. The water potential differs within the peat cores according to hydrostatic equilibrium, with wetter conditions at the deeper layers. This results in a smaller air-filled porosity, and less connected pore space. Microorganisms in deeper layers are therefore 250 exposed to a more oxygen depleted environments. Deeper layers are also characterized by a smaller interface area between the air- and water-filled pores than the shallow layers, which limits the space where the fast heterotrophic respiration is feasible. Furthermore, these layers are more distant from the surface, and pore networks likely show greater tortuosity, which leads to a greater resistance to diffusion of oxygen from the surface to these layers, and vice versa, diffusion of CO2 to the surface.

The two measures ($^{13}CO_2$ produced and $t_{1/2}$) showed great variance that was only partially explained by the main variables 255 in our study (injection depth, injection round, moisture treatment, soil pit), with 31% residual variance in the case of label-derived CO2 and 41% residual variance for $t_{1/2}$ (**Fig S2**). This large unexplained variance likely resulted from difference among



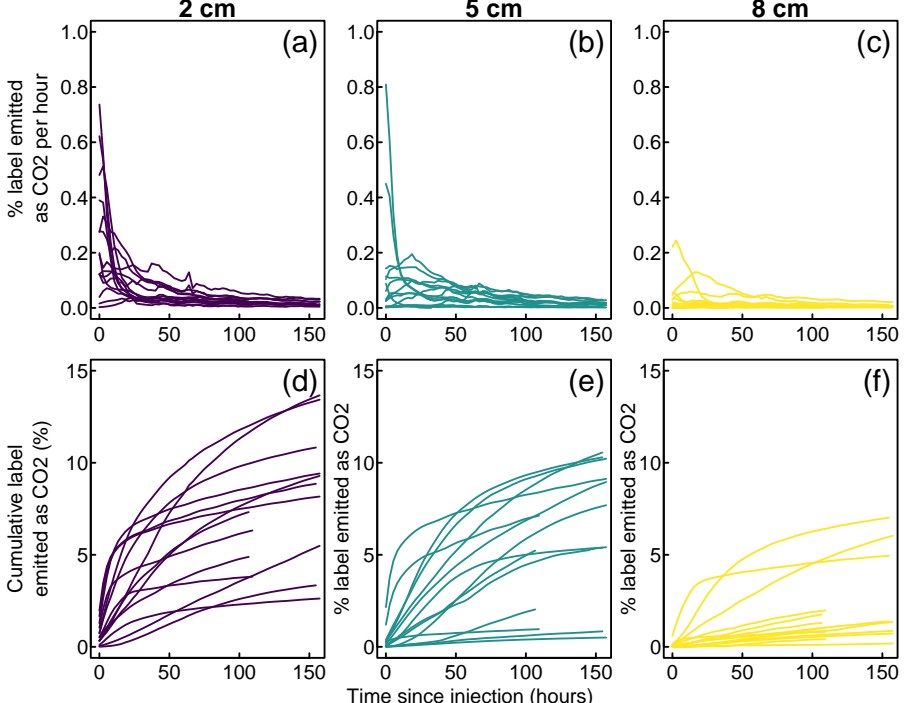

**Figure 5.** Instantaneous (a-c) and cumulative (d-f) emissions of label-derived $CO_2$ grouped by injection depth.

individual soil cores and spatial variability within the cores, and therefore likely represents heterogeneity in peat structure (i.e., pore networks). Injection depth explained 38% of the total emissions of label-derived $CO_2$ and 15% of $t_{1/2}$. Pit-to-pit variation was an important predictor for $t_{1/2}$ (43% variance), but not for the amount of ($^{13}CO_2$ produced. All other independent variables explained less then 10% of the variability.

### 3.4 Label-derived $CH_4$ emissions

The label-derived $CH_4$ emissions of were highly heterogeneous (Figs 7a-7c). Quantitatively, however, the conversion of the injected label to CH4 was very limited, with less than 0.01% of the injected label emitted as methane. We detected $^{13}CH_4$ emissions in both peat cores that showed background (non-labelled) $CH_4$ emission and peat cores that showed no such background emission, but $^{13}CH_4$ emissions increased with higher background emissions (R >0.73, p <0.003, tested separately for each injection depth).

Not all label injections into methane emitting cores resulted in $^{13}CH_4$ emissions. Rather, we found differences between injections into the same peat core, further highlighting within-core heterogeneity. Injections into one of the peat cores (sample 7A), for examples, resulted in situations (i) large $^{13}CH_4$ with little $^{13}CO_2$, (2) emissions of both $^{13}CH_4$ and $^{13}CO_2$, and (3)





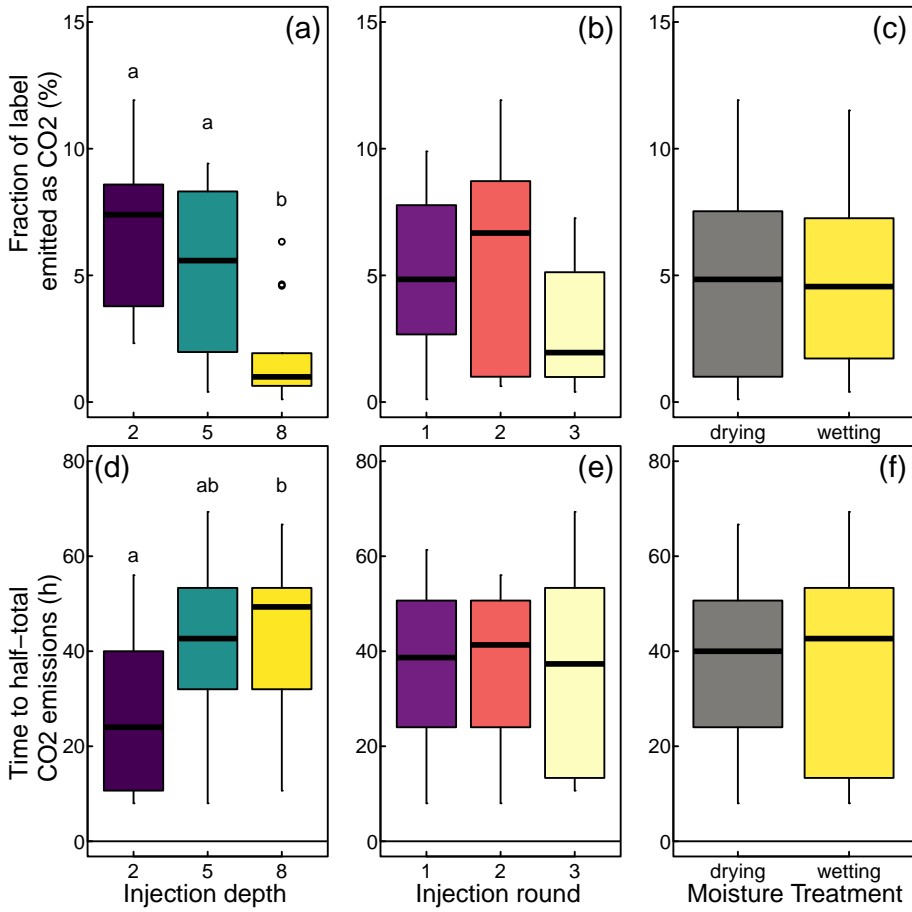

**Figure 6.** Effects of injection depths, injection round, and moisture treatment Fraction of the label emitted as CO2 and delay to half-total emissions.

270   only $^{13}CO_2$ (Fig. 7d). This response was not a simple function of depth – highest 13CH4 emissions were found after injection at intermediate depths, while highest $CO_2$ emissions were found after injection into deepest layer.

Our results thus show that methane production varied both at the scale of tens of centimeters (replicate injections into the same core gave similar responses), and at the cm scale (contrasting results from injections into the same peat core). This highlights the great heterogeneity of peat at sub-site scales. It also indicates presence of methane-generating and non-methane-

275   generating locations within peat cores, likely corresponding to the oxic and anoxic microsites within peat cores (Fan et al., 2014).

The tracing of label-derived $CH_4$ in our study remained associated with some important limitations. First, we measured $^{13}CH_4$ emissions, which differ from $^{13}CH_4$ production. It is likely that the anaerobic pockets where $^{13}CH_4$ is formed are poorly connected to the surface, and that the formed $^{13}CH_4$ may not reach the sample headspace. Indeed, the most $^{13}CH_4$

280   emissions time series (Figs 7a-7c) show continuous emissions over the whole timecourse of the experiment, unlike $^{13}CO_2$



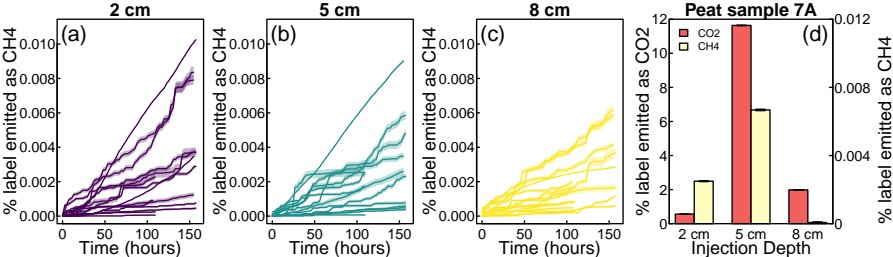

**Figure 7.** Cumulative emissions of label-derived $CH_4$ grouped by injection depth (a-c). Further, a comparison between label-derived $CO_2$ and $CH_4$ emissions after injections at different depth into an example peat core.

emissions which decreased over time 5. This may indicate that acetoclastic methanogenesis occurs slowly than heterotrophic respiration, consistent with the slower nature of anaerobic metabolisms. It may also indicate that $^{13}CH_4$, once formed, reaches the sample surface only slowly. Another limitation is that we cannot exclude that $^{13}CH_4$ formed at the site of label injection is oxidized by methanotrophs prior to reaching the peat surface.

### 3.5 Comparison between porosity, pore network properties, and greenhouse gas emissions

To investigate if the air-filled porosity derived from the $\mu$CT images can explain the heterogeneity of label-derived $CO_2$ emissions between the peat cores and injections, we tested if the average air-filled porosity above the injection depth was correlated to (i) the fractions of the label emitted as $CO_2$ or (ii) the time until half of all emissions had occurred ($t_{1/2}$). This analysis was done separately for each injection depth. We found no correlation between between air-filled porosity and the amount of label-derived $CO_2$ emitted after injections, but greater air-filled porosity was associated more rapid emissions of $^{13}CO_2$ (lower $t_{1/2}$) at all injection depths (Fig. 9).

We also tested for correlations between these measures of and pore network metrics (Fig. **Snn**). Again, we found no correlation between the analysed metrics and the fraction of the label emitted as $CO_2$. The slower release of $^{13}CO_2$ (higehr $t_{1/2}$) were associated with greater clustering coefficients (8cm depth) and betweenness centrality (all depths). These metrics, however, were themselves associated with lower air-filled porosity (Table **Snn**), such that we could not distinguish if differences in $t_{1/2}$ resulted from air-filled porosity *per se* or the properties of the networks described by the network metrics. Clustering coefficients were negatively correlated with air-filled porosity unlike in previous studies (Kiuru et al., 2022a). Greater clustering coefficients, however, indicate a greater network connectivity, which would have the opposite effect on $CO_2$ release time (faster release in more connected networks with greater clustering coefficients). In contrast, higher air-filled porosity indicates a that greater part of the peat receives sufficient oxygen to convert the label to $CO_2$, and that such $CO_2$ can diffuse out of the peat column faster. It is therefore likely that air-filled porosity, not the clustering coefficient, was responsible for the observed correlations. Betweenness centrality indicates the probability that a given pore is part of the shortest connnection between pores at the top and bottom of the peat core. High betweenness centrality indicates that a small number of essential for air transport through the peat cores, an may therefore have contributed to a slower $CO_2$ release.





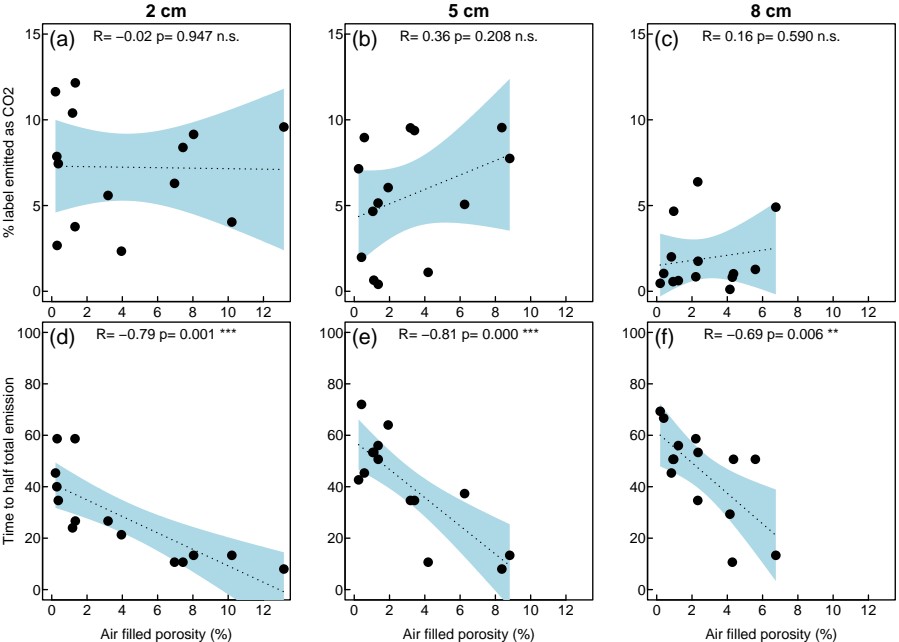

**Figure 8.** Correlations between $\mu$CT-derived air-filled porosity measured from the $\mu$CT images and the percentage of the label that has been emitted as $^{13}CO_2$ after injections at depths of 2 cm (a), 5 cm (b) and 8 cm (c). Further, correlation between air-filled porosity and the time until half of such emission rate had occurred after injections at depths of 2 cm (d), 5 cm (e) and 8 cm (f).

Overall, our results shows that pore network imaging at the rough resolution possible for intact peat cores of the size studied here (10cm) and at (near) native water content provided little additional information to what can be derived from macroscopic measures like injection depth, i.e., distance to surface, and air-filled porosity. Identifying the precise location of the injection in the pore networks might have provided additional information, but was not possible as the wooden injection position markers used in our study were not clearly visible in $\mu$CT images .

**4   Conclusions**

We have established an experimental setup to identify biogeochemical heterogeneity of micro-environments within peat cores that are involved in the production of $CO_2$ and $CH_4$ by combining laboratory-scale manipulation experiments and thorough $\mu$CT imaging of relatively large peat cores. $\mu$CT imaging has been used to study the physical heterogeneity before, but to our best knowledge this was the first attempt to investigate the spatial heterogeneity in biogeochemical transformation rates and 315 gas transport. The highly variable responses to label injections found in our study demonstrate a high heterogeneity of these processes at the centimeter scale.



Our study highlights the significant challenges associated with such an undertaking. The analysis of pore network by $\mu$CT imaging, which allowed us to study pore network architecture at the scale of fractions of millimeters could not remove the remaining uncertainties in what governs the spatial heterogeneity in biogeochemical transformations.

Nevertheless, our study showed that the biogeochemical heterogeneity observed at a scale of centimeters (injection depths) to tens of centimeters (replicate peat cores from the same pit) was as large as the heterogeneity observed at 10s of meters (between pits). Our work thus emphasizes that defining the relevant scale for the investigated processes is of key importance for future studies.

*Code and data availability.* The Python scripts used in the $\mu$CT image processing and calculations are available in GitHub
(https://github.com/pjkiuru/networks_100_microns). The $\mu$CT image and binary image data are available from the (corresponding) authors upon reasonable request. Raw data of the labelling experiment and the code used to process them is available at Zenodo (doi:10.5281/zenodo.11088028).

*Author contributions.* LK, AM, MP, and MR conceptualized the experiment. LK, AM, and MP collected samples in the field. LK conducted the manipulative experiment. LK processed $CO_2$ and $CH_4$ emission data, PK processed and analysed $\mu$CT images. LK conducted the formal analysis. LK and AL wrote the first draft of the manuscript, which was revised based on input from all co-authors.

*Competing interests.* The authors declare no competing interests.

*Acknowledgements.* We thank Tatu Polvinen help constructing the measurement system.





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
