# Peer review of "Exploring micro-scale heterogeneity as a driver of biogeochemical transformations and gas transport in peat"

_EGUsphere, 2024_

## Referee Comment (RC2)

This study performs a lab experiment on peat cores to quantify the effect of peat pore structure on the production and transport of $CH_4$ and $CO_2$. The study is unique because few studies on peat have addressed these processes at these scales. The experimental setup and data analysis are sound and well explained. The study could not provide a definitive linkage between pore structure, gas transport and gas production, but this does not detract from the study's overall value. I have one concern about the study and list several minor corrections that I recommend the authors should address:

It appears that the shrinkage of the peat has affected your analysis of air-filled porosity near the edge of your samples. Although you explain this in the text, the plot in Figure 3f can be misleading. Could you either indicate on Figure 3f where the edge effects begin or subset the data that does not include the edges and replot Figure 3f.

**Minor corrections**
Line 5: remove "the"

Line 12: consider rewriting "air-filled porosity was" as it does not read smoothly

Line 31: add Wright et. al 2018 as a reference for peat pore structure affecting methane emissions in the field
Methane ebullition from subtropical peat: Testing an ebullition model reveals the importance of pore structure, W Wright, JA Ramirez, X Comas - Geophysical Research Letters, 2018

Line 39: instead write "Despite the progress in"

Figure 2: line 2 should be $CH_4$. Throughout all plots, axis labels should be formatted as $CO_2$ and $CH_4$. Remove brackets from axis labels.

Line 108: complete subscript for $V_{mol}$ and $t_{cycle}$, and add symbol for 20 °C

Line 112: complete subscript for $c_{min}$ and $c_{max}$

Line 117: remove bold

Line 119: complete subscript for $R_{ref}$ and $CH_4$

Line 175: typographical error: effecst

Figure 3: add to caption that the air-filled pore space is displayed in black and peat in white, and also add a scale bar

Figure S1: consider adding a legend to easily understand the colors. Marker sizes are difficult to distinguish so make the figure larger on a 2 x 2 layout. Line 3 in caption typographical error: "[he"

Line 207: is this the correct units: μmol hh$^{-1}$ ?

Figure 4: add subscripts and superscripts to y-axis labels. Not sure what you mean by "Letters in panel(b) indicate significant differences between the injection rounds" because no letters appear in the panels. For the methane flux plots, consider adjusting the min and max values of the y-axis for complete visualization of the data (i.e negative fluxes).

Line 213: make sure subscripts and superscripts are applied throughout the remainder of the manuscript (e.g. $CH_4$). Is this correct μmol hh$^{-1}$, if not, please also correct throughout the manuscript.

Line 224: please report the temperature values throughout the duration of the experiment.

Line 226: typographical error: trying

Figure 5: add subscripts to y-axis labels

Figure 6: add subscripts to y-axis labels. Explain in the caption the meaning of the letters (e.g. ab) within the plots.

Line 243: replace til with until

Line 256: remove bold

Line 259: typographical error: ($^{13}CO_2$

Line 262: first sentence reads awkwardly and needs rewriting

Figure 7: subscripts are needed throughout labels and there is no reference to plot d in caption

Line 281: not clear what "time 5" means and replace slowly with slower

Line 291: figure 9 does not exist

Line 292: Fig. Snn?

Line 295: Table Snn?

Line 300: remove "a"

Line 303: "small number of essential..." should be "small number of pores are essential..."?

Figure 8: time on the y-axis label needs units. In the caption provide explanation for asterisks and n.s.

---

## Author Comment (AC1)

***Exploring micro-scale heterogeneity as a driver of biogeochemical transformations and gas transport in peat.***

Kohl et al., in review for Biogeosciences

**Response to reviewers**

*Reviewer comments are printed in italics*, our responses in normal font.

**Reviewer 1**

> *The manuscript explores the impact of micro-scale heterogeneity on biogeochemical processes and gas transport in peat soils using a 13C pulse-chase assay and X-ray microtomographic imaging. While the topic is relevant, several critical issues warrant rejecting this manuscript in its current form.*

We appreciate the reviewer's critical assessment of our work and will improve the manuscript based on their comments in a revision of the manuscript.

Overall, it appears that the reviewer's expectation regarding rigor of our relatively small study by far exceeded our own. We aim to publish this manuscript primarily to document and share our methodological approach, rather than present definitive outcomes. We will therefore adapt the manuscript to better manage the readers expectations. The reviewer's comments also indicate that additional clarity is needed regarding the novelty of our study - a novelty that appeared evident to reviewer 2.

> *Literature Review:*
>
> *The manuscript lacks comprehensive references to existing literature, predominantly citing the authors' previous works. A thorough literature review is essential for contextualizing the study and demonstrating its novelty. The claim that this is the first investigation into spatial heterogeneity in biogeochemical transformation rates appears overstated. Relevant literature includes DOI: 10.1016/j.geoderma.2022.116224, 10.1007/s00374-022-01673-6, 10.1016/j.scitotenv.2023.165192, https://www.ssrn.com/abstract=4092466, 10.1016/j.soilbio.2022.108565, 10.5194/bg-18-1185-2021, 10.1007/s11104-021-04871-7, 10.3389/fenvs.2018.00017, 10.1038/ngeo2963, and 10.1111/gcb.14855.*

The reviewer references a body of literature that is primarily focused on predicting or explaining $N_2O$ production in mineral soils based on the pore architecture. We will incorporate literature suggested by the reviewer, however, we think that these studies are only partially relevant to

our manuscript which deals with methane production in peat soil. $N_2O$ production in mineral soil and $CH_4$ production in peat soil are significantly distinct systems. Most importantly, the relevant pore sizes CH4 production in peat soils tend to be orders of magnitude larger than those relevant for N2O production in mineral soil.

Also, in most of the mentioned studies the linkage between pore architecture and biogeochemical transformation rates remains limited to correlative analysis of pore structure properties vs. whole-sample $N_2O$ production (Kim et al. (2021) plant and soil is a noteworthy exception).

The novelty in our study, in contrast, was our aim to make spatially resolved measurements of actual transformation rates (expressed as the production of 13CH4/13CO2 after injections at distinct depths). The lack of such measurements in the studies mentioned by the reviewer actually supports our claim of novelty.

> *Experimental Design:*
>
> *The study's experimental design lacks clarity or is simply wrong for answering all hypotheses. There were seven replicates for two treatments, but it's unclear how samples were labeled and analyzed at different depths. The division of replicates across three depths at multiple time points compromises the factorial design and diminishes the robustness of replication. A mixed effect model can deal with small sample numbers. However, the nested design, with depths and time points nested within field samples, requires careful statistical handling (need to be included in the error term), which the current sample size may not support adequately.*

In the revised manuscript, we will provide a figure depicting our experimental approach. The limited replication is a consequence of our novel measurement approach, where further increasing the number of replicates was not feasible, thus requiring the nested design. Nevertheless, we think that our replication is appropriate for the study. In the revision, we will clarify this by documenting the mixed effects model in greater detail.

> > *The introduction fails to provide a strong motivation or hypothesis for analyzing different depths. Key parameters are introduced without sufficient context, making many results appear as unstructured data dumps rather than addressing specific research questions.*

The motivation for injecting at different depths was to elucidate the spatial heterogeneity of biogeochemical transformations within peat cores. We will improve the justification of our research questions.

> > *The discussion is weak, lacking depth and failing to integrate findings with existing literature. The conclusion that heterogeneity is significant at the core and within-core scales but not at the stand scale is unsupported by the study design, as the core scale and stand scale are conflated. There are no real replicates on the plot scale.*

With stand scale, we mean pit-to-pit variety within stands, not stand-to-stand variety. We will clarify this in the revision, and we will also improve the integration of our findings with existing literature. Again, this comment indicates that we did not appropriately manage the reader's expectation, in what we intended to publish primarily to document our measurement approach.

*Essential details about the scanning procedure, such as panel size, exposure time, and number of projections, are missing, hindering reproducibility. Additionally, the rationale for cropping images to 90 mm diameter instead of the full 100 mm is unclear, especially considering the impact of fissures and cracks on diffusion, which is evident in the radial porosity analyses.*

We will provide these details in a revision of the manuscript.

*The term "pore network modeling" is misleading. The study employs image analysis algorithms to extract features like porosity and connectivity, but does not engage in actual modeling of pore networks.*

We use the term '*pore network model'* to refer to the mathematical abstraction of uCT images, first into a set of pores and connecting throats (as spheres and tubes), and then further into network graphs. A pore network model is thus a simplified depiction of the actual porespace the same way that a map is a simplified model of the actual landscape. This is a common use of the term in the scientific literature.

Using these graphs allows us to use the mathematical tools provided by network theory to analyse pore networks This would not be possible with depictions of the pore space in its actual complexity. We therefore argue that pore network analysis is the appropriate term in this study.

*Minor Comments*

*There are several language mistakes / missing words*

*Abstract: The sentence "Greater peat air-filled porosity was and pore network metrics could not explain the fraction of label converted to CO2, but greater porosity as well as higher clustering coefficients and betweenness centrality were associated with slower CO2 emissions" needs correction. Clustering coefficients and betweenness centrality should be introduced for reader comprehension.*

*Introduction: The first two sentences are misplaced. Begin with the broader relevance of the study.*

*Line 18: Contrary to the authors' claim, there is a growing body of literature using X-ray CT and other methods to explore pore heterogeneity in soil functions.*

*Line 27: Replace with "anaerobic."*

*Line 172: Clarify that certain parameters are critical for transport properties but are not transport properties themselves. Provide references.*

*Figures: Ensure all y-axis titles in Figures (e.g., Fig. 2, Figs. 4-6) have correct subscripts. The physics of porous media should be correctly attributed (Line 36). Consider moving Fig. 2 to supplementary information.*

*Statistical Analysis: Include a reference for the mixed effect model and details on testing assumptions. The p-value in Fig. 8(e) should be clarified.*

We address these comments in a revision of the manuscript.

**Reviewer 2**

*This study performs a lab experiment on peat cores to quantify the effect of peat pore structure on the production and transport of CH4 and CO2. The study is unique because few studies on peat have addressed these processes at these scales. The experimental setup and data analysis are sound and well explained. The study could not provide a definitive linkage between pore structure, gas transport and gas production, but this does not detract from the study's overall value. I have one concern about the study and list several minor corrections that I recommend the authors should address:*

We thank Reviewer 2 for their positive feedback on our work.

*It appears that the shrinkage of the peat has affected your analysis of air-filled porosity near them edge of your samples. Although you explain this in the text, the plot in Figure 3f can be misleading. Could you either indicate on Figure 3f where the edge effects begin or subset them data that does not include the edges and replot Figure 3f.*

Shrinkage was indeed a minor issue during our study. We therefore excluded the horizontal edges from further analysis. We will update Figure 3f to indicate this more clearly.

*Minor corrections*
*Line 5: remove "the"*
*Line 12: consider rewriting "air-filled porosity was" as it does not read smoothly*
*Line 31: add Wright et. al 2018 as a reference for peat pore structure affecting methane emissions in the field*
*Methane ebullition from subtropical peat: Testing an ebullition model reveals the importance of*
*pore structure, W Wright, JA Ramirez, X Comas - Geophysical Research Letters, 2018*

*Line 39: instead write "Despite the progress in"*

*Figure 2: line 2 should be CH4. Throughout all plots, axis labels should be formatted as CO2 and*

*CH4. Remove brackets from axis labels.*

*Line 108: complete subscript for Vmol and tcycle, and add symbol for 20 °C*

*Line 112: complete subscript for cmin and cmax*

*Line 117: remove bold*

*Line 119: complete subscript for Rref and CH4*

*Line 175: typographical error: effecst*

*Figure 3: add to caption that the air-filled pore space is displayed in black and peat in white, and*

*also add a scale bar*

*Figure S1: consider adding a legend to easily understand the colors. Marker sizes are difficult to*

*distinguish so make the figure larger on a 2 x 2 layout. Line 3 in caption typographical error:*

*"[he"*

*Line 207: is this the correct units: μmol hh-1 ?*

*Figure 4: add subscripts and superscripts to y-axis labels. Not sure what you mean by "Letters*

*in panel(b) indicate significant differences between the injection rounds" because no letters*

*appear in the panels. For the methane flux plots, consider adjusting the min and max values of*

*the y-axis for complete visualization of the data (i.e negative fluxes).*

*Line 213: make sure subscripts and superscripts are applied throughout the remainder of the*

*manuscript (e.g. CH4). Is this correct μmol hh-1, if not, please also correct throughout the*

*manuscript.*

*Line 224: please report the temperature values throughout the duration of the experiment.*

*Line 226: typographical error: trying*

*Figure 5: add subscripts to y-axis labels*

*Figure 6: add subscripts to y-axis labels. Explain in the caption the meaning of the letters (e.g.*

*ab) within the plots.*

*Line 243: replace til with until*

*Line 256: remove bold*

*Line 259: typographical error: (13CO2*

*Line 262: first sentence reads awkwardly and needs rewriting*

*Figure 7: subscripts are needed throughout labels and there is no reference to plot d in caption*

*Line 281: not clear what "time 5" means and replace slowly with slower*
*Line 291: figure 9 does not exist*
*Line 292: Fig. Snn?*
*Line 295: Table Snn?*
*Line 300: remove "a"*
*Line 303: "small number of essential..." should be "small number of pores are*
*essential..."?*
*Figure 8: time on the y-axis label needs units. In the caption provide explanation for*
*asterisks*
*and n.s.*

We address these comments in a revision of the manuscript.

---

## Author Response (AR1)

Dear Steven Bouillon,

Please find attached our revised manuscript "*Exploring micro-scale heterogeneity as a driver of biogeochemical transformations and gas transport in peat.*". Based on the reviewer's feedback, we have re-organized the manuscript to provide a separate discussion section, which now goes to substantially greater depths than in our initial submission. We paid particular attention to better incorporating our work in the body of existing literature, and to better manage the readers' expectations in a manuscript that we conceived primarily as a report on a new measurement approach. We think that this revision has resulted in a much-strengthened manuscript.

Please find our detailed response to the reviewer's comments below. *Reviewer comments are printed in italics*, our responses in normal font.

sincerely,
Lukas Kohl on behalf of all co-authors.

**Reviewer 1**

*The manuscript explores the impact of micro-scale heterogeneity on biogeochemical processes and gas transport in peat soils using a 13C pulse-chase assay and X-ray microtomographic imaging. While the topic is relevant, several critical issues warrant rejecting this manuscript in its current form.*

We appreciate the reviewer's critical assessment of our work which, by identifying crucial shortcomings, have helped us to improve the manuscript in our revision.

*Literature Review:*
*The manuscript lacks comprehensive references to existing literature, predominantly citing the authors' previous works. A thorough literature review is essential for contextualizing the study and demonstrating its novelty. The claim that this is the first investigation into spatial heterogeneity in biogeochemical transformation rates appears overstated. Relevant literature includes DOI: 10.1016/j.geoderma.2022.116224, 10.1007/s00374-022-01673-6, 10.1016/j.scitotenv.2023.165192, https://www.ssrn.com/abstract=4092466, 10.1016/j.soilbio.2022.108565, 10.5194/bg-18-1185-2021, 10.1007/s11104-021-04871-7, 10.3389/fenvs.2018.00017, 10.1038/ngeo2963, and 10.1111/gcb.14855.*

**Revised as requested.** This comment highlights the need to clarify the novelty of our method, and better place it in the context of existing work. In response we have thoroughly revised the introduction and discussion sections to better express this novelty, and to place it in the context of existing literature, and to better manage the readers' expectations.

Our revision now contains substantially expanded both introduction and discussion sections. The literature mentioned by the reviewer studies denitrification in mineral soils, while our manuscript is focused on methane and $CO_2$ production in peat soils. In our revision, we have aimed to cover the existing literature in both peat and mineral soil contexts instead of primarily focusing only on peat soils (L42-50). We also emphasize that most work so far has focused on mineral soils while peat soils remain under-covered.

Further, we have carefully reworded our main statements of novelty: We now emphasize that we aimed to develop a method to directly quantify (potential) biogeochemical process rates at a given location in a soil core, which we differentiate from previous methods to indirectly study the biogeochemistry (e.g. by comparison of pore network properties with gas emissions from soil cores). We also clarify the difference between our approach and microprobe-based measurements as well as zygometry (L48-49).

*Experimental Design:*

*The study's experimental design lacks clarity or is simply wrong for answering all hypotheses. There were seven replicates for two treatments, but it's unclear how samples were labeled and analyzed at different depths. The division of replicates across three depths at multiple time points compromises the factorial design and diminishes the robustness of replication. A mixed effect model can deal with small sample numbers. However, the nested design, with depths and time points nested within field samples, requires careful statistical handling (need to be included in the error term), which the current sample size may not support adequately.*

**Clarified as requested** (L191-205). The design is appropriate for our study, the nested design was carefully handled during statistical analysis, and all analyses are adequately supported by the sample size.

For label-derived $CO_2$ and $t_{1/2}$ (i.e., the result now presented in Fig 7-8), our primary interest was to compare two treatments (fixed effects): moisture treatment (2 levels) and injection depth (3 levels). Other effects, including pit-to-pit and core-to-core variability as well as the order in which the labelled substrate was injected at different depths (see below) were included in the analysis as random effects (pit-to-pit and core-to-core) or fixed effects (order of injections) to control for their potential effects– these are the 'error terms' mentioned by the reviewer. Note that our analysis shows now significant effects of the order of injections ('injection round'). The revised manuscript now includes a more detailed description of the statistical treatment.

The revised manuscript now also includes the results of additional statistical analysis, i.e., the separate analysis of wetting and drying peat cores (Fig 8). We include this analysis because (a) it is somewhat simpler to establish validity and (b) to explore interaction effects of injection depths and moisture treatment by comparing the injection depth effects observed in the drying and wetting subsets (Fig 8).

For moisture treatment effects, the adequacy of replication is trivial: We follow a split-plot design (each soil pit is a plot, each core a subplot) with paired measurements (the two cores from each pit were always treated equally: they received injections at different depths in the same order). All individual injections within a core are pseudo-replicates, resulting in n=7 for each treatment. (effects of injection depths and injection order are controlled for by other fixed variables in the model).

For effects of injection depths, adequacy is best demonstrated within the drying or wetting subsets. Here, we have 21 replicate injections into 7 cores. We have injection depth (n=3), injection order (which injection was conducted first; n=3), and core (n=7) are potential predictors. Core and injection order were included in the model (as random and fixed effects, respectively) for control but had no significant effects on the dependent variables (label-derived $CO_2$ and $t_{1/2}$). We therefore effectively compare the effects of injection depth in 21 replicate injections in a factorial repeated-measurements experiment (i.e., effective n=7 for each depth). The larger model including both treatments follows a analogue logic.

For background $CO_2$ emissions (now in Fig. 5), our model is identical, except that we treat injection round as our independent variable of interest and injection depths as a confounding factor included for control.

Note that we removed our analysis of pit-to-pit variability, which was less well supported, for increased clarity regarding our key findings.

*The introduction fails to provide a strong motivation or hypothesis for analyzing different depths. Key parameters are introduced without sufficient context, making many results appear as unstructured data dumps rather than addressing specific research questions.*

**Revised as requested.** We thoroughly revised the introduction to better introduce and justify our research questions.

*The discussion is weak, lacking depth and failing to integrate findings with existing literature.*

**Revised as requested.** In the revised manuscript, we split the results and discussion sections. The discussion section was completely rewritten.

> *The conclusion that heterogeneity is significant at the core and within-core scales but not at the stand scale is unsupported by the study design, as the core scale and stand scale are conflated. There are no real replicates on the plot scale.*

**This is not correct.** Within-stand replication is represented by differences between soil pits, which were located in >30m distance from each other and therefore cover heterogeneity within forest stnads. We did not draw conclusions about stand-to-stand heterogeneity. **Note that this aspect of our discussion was de-emphasized in the revised manuscript.**

> *Essential details about the scanning procedure, such as panel size, exposure time, and number of projections, are missing, hindering reproducibility. Additionally, the rationale for cropping images to 90 mm diameter instead of the full 100 mm is unclear, especially considering the impact of fissures and cracks on diffusion, which is evident in the radial porosity analyses.*

**Clarified as requested.** These details are now provided in L144-152.

> *The term "pore network modeling" is misleading. The study employs image analysis algorithms to extract features like porosity and connectivity, but does not engage in actual modeling of pore networks.*

**Not changed.** We use the term '*pore network model*' to refer to the mathematical abstraction of uCT images, first into a set of pores and connecting throats (as spheres and tubes), and then further into network graphs. A pore network model is thus a simplified depiction of the actual pore space the same way that a map is a simplified model of the actual landscape.

There is some variation or ambiguity in the terminology in the literature (pore network model / pore network modelling / pore network analysis). Some sources use the term '*pore network model*' to describe a structural model of the pore space consisting of pore bodies and pore throats while other sources use the term only for simulations performed in the network. We think that our use of '*pore network model*' therefore falls within the range of use employed by other sources. **This is, however, ultimately an editorial decision, and we are happy to replace '*pore network model*' with the '*network representation of the pore space*' if requested by the editor.**

> *Minor Comments*
> *There are several language mistakes / missing words*
> *Abstract: The sentence "Greater peat air-filled porosity was and pore network metrics could not explain the fraction of label converted to CO2, but greater porosity as well as higher clustering coefficients and betweenness centrality were associated with slower CO2 emissions" needs correction. Clustering coefficients and betweenness centrality should be introduced for reader comprehension.*

**Fixed.** Clustering coefficient and betweenness centrality are explained in the article body (L187-190).

> *Introduction: The first two sentences are misplaced. Begin with the broader relevance of the study.*

**Not changed.** We deliberately start the article by stating the knowledge gap front-up to engage the reader. We consider this a legitimate stylistic choice.

> *Line 18: Contrary to the authors' claim, there is a growing body of literature using X-ray CT and other methods to explore pore heterogeneity in soil functions.*

**Changed as requested.** We added the following sentence to clarify our knowledge gap:
*"This is especially the case for peat soils, which possess complex pore structures distinct from mineral soils and which remain comparatively understudied compared minerals agricultural soils (McCarter et al., 2020)."* (L24-26)

*Line 27: Replace with "anaerobic."*
**Changed as requested** (L34).

*Line 172: Clarify that certain parameters are critical for transport properties but are not transport properties themselves. Provide references.*
**Changed as requested** (L189-190).

*Figures: Ensure all y-axis titles in Figures (e.g., Fig. 2, Figs. 4-6) have correct subscripts. The physics of porous media should be correctly attributed (Line 36). Consider moving Fig. 2 to supplementary information. Statistical Analysis: Include a reference for the mixed effect model and details on testing assumptions. The p-value in Fig. 8(e) should be clarified.*
**Changed as requested.** We decided to keep Fig. 2 in the main manuscript.

**Reviewer 2**

*This study performs a lab experiment on peat cores to quantify the effect of peat pore structure on the production and transport of CH4 and CO2. The study is unique because few studies on peat have addressed these processes at these scales. The experimental setup and data analysis are sound and well explained. The study could not provide a definitive linkage between pore structure, gas transport and gas production, but this does not detract from the study's overall value. I have one concern about the study and list several minor corrections that I recommend the authors should address:*

We thank Reviewer 2 for their positive feedback on our work.

*It appears that the shrinkage of the peat has affected your analysis of air-filled porosity near them edge of your samples. Although you explain this in the text, the plot in Figure 3f can be misleading. Could you either indicate on Figure 3f where the edge effects begin or subset them data that does not include the edges and replot Figure 3f.*

**Changed as requested.** We added a dashed line to Fig 3f to indicate the limit of the analayzed domain.

*Minor corrections*
*Line 5: remove "the"*
**Changed as requested** (L5).

*Line 12: consider rewriting "air-filled porosity was" as it does not read smoothly*
**Changed as requested** (L14).

*Line 31: add Wright et. al 2018 as a reference for peat pore structure affecting methane emissions in the field Methane ebullition from subtropical peat: Testing an ebullition model reveals the importance of pore structure, W Wright, JA Ramirez, X Comas - Geophysical Research Letters, 2018*
**Changed as requested** (L39).

*Line 39: instead write "Despite the progress in"*
**Changed as requested** (L40).

*Figure 2: line 2 should be CH4. Throughout all plots, axis labels should be formatted as CO2 and CH4. Remove brackets from axis labels*

**Changed as requested.** Brackets were retained, this is a common shorthand for indicating concentrations.

*Line 108: complete subscript for Vmol and tcycle, and add symbol for 20 °C*
**Changed as requested** (L117).

*Line 112: complete subscript for cmin and cmax*
**Sentence removed.** We re-wrote this section to correct a minor math error.

*Line 117: remove bold*
**Changed as requested** (L121).

*Line 119: complete subscript for Rref and CH4*
**Changed as requested** (L124).

*Line 175: typographical error: effecst*
**Changed as requested** (L5).

*Figure 3: add to caption that the air-filled pore space is displayed in black and peat in white, and also add a scale bar*
**Changed as requested.**

*Figure S1: consider adding a legend to easily understand the colors. Marker sizes are difficult to distinguish so make the figure larger on a 2 x 2 layout. Line 3 in caption typographical error: "[he"*
**Changed as requested.**

*Line 207: is this the correct units: μmol hh-1 ?*
**Clarified as requested**  (L241). Corrected to umol h$^{-1}$.

*Figure 4: add subscripts and superscripts to y-axis labels. Not sure what you mean by "Letters in panel(b) indicate significant differences between the injection rounds" because no letters appear in the panels. For the methane flux plots, consider adjusting the min and max values of the y-axis for complete visualization of the data (i.e negative fluxes).*
**Changed as requested.** The reference to letters was removed because we found no significant differences in the data presented in this plot.

*Line 213: make sure subscripts and superscripts are applied throughout the remainder of the manuscript (e.g. CH4). Is this correct μmol hh-1, if not, please also correct throughout the manuscript.*
**Changed as requested.**

*Line 224: please report the temperature values throughout the duration of the experiment.*
**Clarified as requested** (L338).

*Line 226: typographical error: trying*
**Changed as requested** (L343; now drying).

*Figure 5: add subscripts to y-axis labels*
**Changed as requested.**

*Figure 6: add subscripts to y-axis labels. Explain in the caption the meaning of the letters (e.g. ab) within the plots.*

**Changed as requested.** The caption now includes a statement that 'Letters indicate significant difference between groups'.

*Line 243: replace til with until*

**Changed as requested** (L141).

*Line 256: remove bold*

**Section removed**.

*Line 259: typographical error: (13CO2*

**Section removed.**

*Line 262: first sentence reads awkwardly and needs rewriting*

**Changed as requested.** Corrected to "*The label-derived CH$_4$ emissions showed highly variable responses to the individual label injections*" (L277).

*Figure 7: subscripts are needed throughout labels and there is no reference to plot d in caption*

**Changed as requested.**

*Line 281: not clear what "time 5" means and replace slowly with slower*

**Clarified as requested** (L424).

*Line 291: figure 9 does not exist*

**Clarified as requested** (reference to Fig. 10).

*Line 292: Fig. Snn?*

**Clarified as requested** (L433).

*Line 295: Table Snn?*

**Clarified as requested** (L436).

*Line 300: remove "a"*

**Changed as requested** (L440).

*Line 303: "small number of essential..." should be "small number of pores are essential..."?*

**Changed as requested** (L444).

*Figure 8: time on the y-axis label needs units. In the caption provide explanation for asterisks and n.s.*

**Changed as requested.**